# Multilayer Bolometric Structures for Efficient Wideband Communication Signal Reception

**DOI:** 10.3390/nano14020141

**Published:** 2024-01-08

**Authors:** Anna V. Bogatskaya, Nikolay V. Klenov, Alexander M. Popov, Andrey E. Schegolev, Pavel A. Titovets, Maxim V. Tereshonok, Dmitry S. Yakovlev

**Affiliations:** 1Faculty of Physics, Lomonosov Moscow State University, 119991 Moscow, Russia; abogatskaya@mics.msu.su (A.V.B.); nvklenov@mail.ru (N.V.K.); alexander.m.popov@gmail.com (A.M.P.); 2P. N. Lebedev Physical Institute, Russian Academy of Sciences, 119991 Moscow, Russia; 3Superconducting Quantum Computing Lab, Russian Quantum Center, Skolkovo, 143025 Moscow, Russia; 4D. V. Skobeltsyn Institute of Nuclear Physics, Lomonosov Moscow State University, 119991 Moscow, Russia; a.e.schegolev@pn.sinp.msu.ru; 5Science and Research Department, Moscow Technical University of Communication and Informatics, 111024 Moscow, Russia; paveltitovec@mail.ru (P.A.T.); m.v.tereshonok@mtuci.ru (M.V.T.); 6Laboratoire de Physique et d’Etude des Matériaux, ESPCI Paris, CNRS, PSL University, 75005 Paris, France

**Keywords:** wideband communications, bolometer, dielectric resonator, absorption of electromagnetic radiation in plasma

## Abstract

It is known that the dielectric layer (resonator) located behind the conducting plate of the bolometer system can significantly increase its sensitivity near the resonance frequencies. In this paper, the possibility of receiving broadband electromagnetic signals in a multilayer bolometric meta-material made of alternating conducting (e.g., silicon semiconductor) and dielectric layers is demonstrated both experimentally and numerically. It is shown that such a multilayer structure acts as a lattice of resonators and can significantly increase the width of the frequency band of efficient electromagnetic energy absorption. The parameters of the dielectric and semiconductor layers determine the frequency bands. Numerical modeling of the effect has been carried out under the conditions of our experiment. The numerical results show acceptable qualitative agreement with the experimental data. This study develops the previously proposed technique of resonant absorption of electromagnetic signals in bolometric structures.

## 1. Introduction

The demand for communication systems is growing rapidly, prompting engineers and researchers to explore more and more frequency bands. At the same time, it is imperative to address the issue of electromagnetic spectrum pollution [1]. The solution to these two conflicting challenges lies in using low-level radio signal radiation and receivers with quantum sensitivity levels. Devices operating at the single-photon level play a key role in several applications, including boson sampling [2], quantum simulators [3], and linear optical quantum computing [4].

Detectors with quantum sensitivity are currently standard for photons, but only in the optical frequency (wavelength) range [5,6]. Superconducting bolometric detectors are commonly used for infrared light (e.g., at the important vital wavelength of 1550 nm). In this case, the energy spectrum of the system is such that even a single quantum of radiation puts a thin and narrow superconducting strip into a resistive state (changes the state of the electronic collective). This produces a voltage pulse at the detector output [7,8,9]. In contrast to visible and infrared wavelengths, developing devices in the microwave frequency range pose more significant challenges, mainly due to the photon energy that is lower by four orders of magnitude. Detectors based on artificial quantum systems have recently been demonstrated [10,11,12,13]. However, they suffer from several fundamental drawbacks in performing their function: narrow bandwidth, the need to fine-tune parameters during operation, and scalability. This makes them less suitable for practical applications. An alternative approach is to use narrow-bandwidth superconductors to detect microwaves via a pair-breaking mechanism using microwave surface plasmon (SP) sensors. These include kinetic inductance detectors (KIDs) [14,15], superconducting nanowire single-photon detectors [16,17], and transition edge sensors [18,19]. Existing SP-based devices operate in the mid-infrared (IR) region [20,21]. A practical approach to overcome these challenges is the careful selection of the detector material [22,23], which can include high-temperature superconductors (HTSC) [24,25,26]. The implementation of increasingly complex heterostructures has enabled the development of ultra-sensitive detectors for longer wavelengths and lower photon energies [27,28,29,30,31,32,33,34,35,36,37,38]. Quantum sensitivity has also been achieved for radiofrequency signals [39,40].

In our previous article [41], we considered utilizing well-known resonance effects [42,43,44,45,46,47] to increase the efficiency of any bolometric detector. By choosing the material and thickness of the dielectric substrate that acts as a resonator so that a half-integer number of standing wavelengths can fit within this size, we can significantly increase the probability of detecting individual photons.

It is worth noting that the proposed versions of the high-sensitivity bolometric detectors are resonant. This limits the effective adequate receiving bandwidth. However, the solution is to create broad spectral zones by adding several conductive and dielectric layers (i.e., by constructing anti-reflection nanomaterials). Broad spectral zones of electromagnetic energy absorption make it possible to detect broadband signals. Of course, this solution has a drawback: each layer absorbs and reflects signal energy, reducing the overall energy efficiency of the device.

The main objective of the present study is to demonstrate the possibility of broadband electromagnetic signal detection with high efficiency based on the multilayer bolometric structure of conducting and non-conducting layers. Such a receiver should meet the requirements of modern communication applications [48].

This paper unveils the outcomes of comprehensive theoretical and experimental inquiries into the wideband absorption of gigahertz waves within a structure comprising alternating silicon semiconductor and dielectric layers. The experimental findings showcase the possibility of a reasonable balance between bandwidth and efficiency. Moreover, our investigations illuminate a pathway to further enhancing detector efficiency by transitioning from a “resonator-as-substrate” layer sequence to an “anti-reflection coating” configuration.

## 2. Modelling of Electromagnetic Wave Propagation in Periodic Heterostructures

It is well known that the energy levels of a quantum system split when moving from a potential well to a pair of connected identical wells. In Dirac comb-type systems with many potential wells, typically relatively broad energy zones of the total energy levels appear to exist. The quantum mechanical analogy tells us that it is possible to obtain relatively wide absorption bands instead of narrow resonance peaks by replacing one set of “bolometer + dielectric resonator” layers with several such sets [49,50,51,52,53,54,55]. Such a structure with allowed and forbidden spectral intervals for electromagnetic wave propagation can be considered as a photon crystal.

Previous studies have shown that it is challenging for the radio frequency range to consider a pair of bolometric and dielectric layers as a potential pit and a high barrier. It is all the more important to verify the assumption made from general considerations by an accurate calculation for electromagnetic fields and to compare the results of the analysis with experimental data.

Our theoretical investigations of the propagation of the electromagnetic wave through the “semiconductor-dielectric” heterostructure embedded in the rectangular waveguide (we plan to consider a sequence of regions with different transparency) are similar to those applied in [41]. We assume that only the lowest *H*_10_ mode propagates in the waveguide: the electric field vector has only a transverse component, while the magnetic field vector is characterized by both transverse and longitudinal components [44]. The analysis of the electromagnetic field propagation should be carried out using the Helmholtz equation for the electric field strength $E_{\omega }$ [43,44]:(1)$$\frac{\partial^{2}E_{\omega }}{\partial z^{2}}+k_{⊥}^{2}E_{\omega }=0.$$

Here, the *z*-axis is directed along the waveguide, ω is the angular frequency of the radiation, and
(2)$$k_{⊥}^{2}=\frac{\omega^{2}}{c^{2}}\left(\varepsilon \left(z\right)-\frac{\omega_{c}^{2}}{\omega^{2}\varepsilon \left(z\right)}\right)$$
is the transverse wave number, which depends on the permittivity profile along the z-axis $\varepsilon \left(z\right)$; c is the speed of light in vacuum, and $\omega_{c}$ is the cut-off frequency in the waveguide. For the empty waveguide, it is given by the expression $\omega_{c}=\pi c/l$, where l is the internal transverse size of the waveguide. This cut-off frequency represents the minimum value of the wave frequency that can be transmitted in the given waveguide. 

In this paper, we consider different configurations of the sequence of dielectric and conductive layers, including the two-layer and four-layer periodic structures. Specifically, the permittivity profiles of the studied heterostructures are written in the following forms (the electromagnetic wave is incident from the right):(3)$$\varepsilon \left(z\right)=\left\{\begin{array}{c} \varepsilon_{air},z\leq 0,\\ \varepsilon_{p},0<z\leq d_{1},\\ \varepsilon_{d},d_{1}<z\leq a+d_{1},\\ \varepsilon_{air},a+d_{1}<L_{max} \end{array}\right.$$
(4)$$\varepsilon \left(z\right)=\left\{\begin{array}{c} \varepsilon_{air},\;\;z\leq 0,\\ \varepsilon_{p},\;\;0<z\leq d_{1},\\ \varepsilon_{d},\;\;d_{1}\leq z\leq a+d_{1},\\ \varepsilon_{p},\;\;a+d_{1}<z\leq a+d_{1}{+d}_{2},\\ \varepsilon_{d},a+d_{1}{+d}_{2}<z\leq 2a+d_{1}{+d}_{2},\\ \varepsilon_{air},\;\;2a+d_{1}+d_{2}<L_{max}\\ . \end{array}\right.$$
(5)$$\varepsilon \left(z\right)=\left\{\begin{array}{c} \varepsilon_{air},\;\;z\leq 0,\\ \varepsilon_{p},\;\;0<z\leq d_{2},\\ \varepsilon_{d},\;\;d_{2}\leq z\leq a+d_{2},\\ \varepsilon_{p},\;\;a+d_{2}<z\leq a+d_{1}{+d}_{2},\\ \varepsilon_{d},a+d_{1}{+d}_{2}<z\leq 2a+d_{1}{+d}_{2},\\ \varepsilon_{air},\;\;2a+d_{1}+d_{2}<L_{max}\\ . \end{array}\right.$$

Here, we use two thin doped semiconductor plates (“1” and “2”) arranged in different ways between two dielectric plates, as well as the single semiconductor plate “1” placed behind the dielectric plate (the case of “anti-reflection coating”). $\varepsilon_{air}≅1$ is the permittivity of the air in the waveguide, $\varepsilon_{d}=\varepsilon_{d0}+i\times tan{\;\delta }_{\varepsilon }$ is the permittivity of the dielectric plate (this value also includes the losses described by the term$\;tan{\;\delta }_{\varepsilon }$), a=1.3 cm is the width of the dielectric plate, and $d_{1}=0.4$ cm and $d_{2}=0.35$ cm are the widths of the doped silicon semiconductor plates with the corresponding electrical resistivity values $\rho_{1}=$ 47 ± 3 and $\rho_{2}=$ 21 ± 3 Ω∙cm. The permittivity of the semiconductor plates $\varepsilon_{p}$ was introduced by the Drude formula [45]: (6)$$\varepsilon_{p}=\varepsilon_{u}-\frac{\omega_{p}^{2}}{\omega^{2}+\nu^{2}}+i\frac{\omega_{p}^{2}\nu }{\left(\omega^{2}+\nu^{2}\right)\omega }.$$

Here, $\varepsilon_{u}\approx 12$ is the permittivity of the undoped silicon semiconductor layer, $\omega_{p}=\sqrt{4\pi e^{2}n_{e}/m^{*}}$, ν are the plasma frequency and the transport collisional frequency, where we set the value of $\nu =1.25\times 10^{12}$ s^−1^ similar to [41]; $n_{e}$ is the density of n-type charge carriers, which is equal to $1.02\times 10^{14}$ cm^−3^ for the plate “1” and $2.3\times 10^{14}$ cm^−3^ for the plate “2”; $m^{*}≅1.08\;m$ is the electron effective mass in the silicon (*m* is an electron mass). The boundary conditions and the numerical procedure for the integration of the wave Equation (1) were similar to those given in [41,44]. The software package “Wolfram Mathematica” was used to perform numerical simulations.

## 3. Experimental Setup and Measurements; Results and Discussion

### 3.1. Experimental Setup and Measurements

We have developed an experimental setup for the comparative analysis of experimental and numerical results, as shown in Figure 1. The central element of the experimental setup was a closed-type waveguide system with two ports for the supply and detection of electromagnetic radiation. To build a closed-type waveguide system, we used two coaxial waveguide junctions (CWJ) and a waveguide, the length of which is 50 cm; the internal size of the CWJ and the waveguide is 5.8 × 2.5 cm, and the operating frequency range in the single-mode radio wave propagation regime is from 2.5 to 6 GHz. A vector network analyzer (VNA), S5085 [48], was used as a source and detector of electromagnetic radiation, with two CWJs connected to its ports. The VNA was calibrated with the coaxial cables (CCs) before the output of the coaxial cable connector using the 6550F18-F calibration standard kit (full two-port calibration) to eliminate the influence of the CCs on the input and detected electromagnetic radiation.

We measured the power of the electromagnetic radiation passing through the heterostructure (dielectric–semiconductor–dielectric–semiconductor) at the output of the CWJ port (left in Figure 1). The CWJ electromagnetic radiation was fed to another port (right in Figure 1). Fluctuations displayed on the VNA screen are associated with re-reflections of electromagnetic radiation within the waveguide and the occurrence of standing waves within the waveguide system.

The heterostructure consisted of:

(3) Two dielectric plates (polymer dielectric ST-16 with geometric dimensions of 5.8 × 2.5 × 1.3 cm, a dielectric constant $\varepsilon_{d0}=16.1$, and a loss tangent $tan{\;\delta }_{\varepsilon }=2\times 10^{-3}$) as described in (3);

(4) Semiconductor wafer (silicon doped with phosphorus having geometric dimensions of 5.8 × 2.5 × 0.4 cm and an average resistivity of 49.5 Ω × cm) as described in (4);

(5) Semiconductor wafer (silicon doped with phosphorus having geometric dimensions of 5.8 × 2.5 × 0.35 cm and an average resistivity of 17 Ω × cm) as described in (5).

It should be noted that our waveguide can only operate in the single-mode regime in the band f=2.6−6 GHz, as 3 GHz is close to the limit and the multimode regime is found above 6 GHz.

### 3.2. Results and Discussion

We initiate our analysis of the resonant absorption effect by considering a structure composed of a doped semiconductor layer positioned behind the dielectric plate, as depicted in expression (3). The experimental and simulation data of the radiation absorption by such a structure as a function of its frequency are shown in Figure 2a. It is evident that the absorption experiences resonance in the presence of the dielectric plate approximately at the linear frequency of f=ω/2π≈4.3 GHz. Figure 2b illustrates that the presence of a dielectric rearranges the field of the electromagnetic wave in the vicinity of the semiconductor plate, thereby leading to an augmented level of absorption within it. The field modulus distribution inside the resonant structure clearly shows that the resonance corresponds to the condition when an integer number of half-wavelengths fits along the length of the resonator (three half-wavelengths). 

Another part of the work is devoted to investigating the possibilities of controlling the fraction of the signal absorbed as well as the absorption bandwidth, which is important for the detection of broadband signals. To this end, we have studied the characteristics of the resonant absorption of radiation in the presence of four-layer structures whose dielectric permittivity is described by Formulas (4) and (5). The measured and simulated radiation absorption data for structure (4) are shown in Figure 3. A significant increase in the fraction of the absorbed signal can be observed compared to the previous case, as well as a slight shift in the resonance frequency (Figure 3a). In this case, the resonance condition (three half-wavelengths) is fulfilled independently for both dielectric plates (see Figure 3b). By contrast, the dielectric plate located between the two semiconductors acts as a “booster”, providing effective penetration (and absorption) of the wave field inside the far (concerning the incident wave) semiconductor plate (plate “1”). Nevertheless, it is clear from Figure 3b that the value of the field distribution in the left dielectric plate is considerably lower than in the right one, so the absorption is mainly determined by the semiconductor plate “2”, which is placed closer to the incident wave. Of note, this version of “anti-reflection coating” with good matching of “distributed detector impedance” to “open space impedance” provides a noticeable increase in absorption.

A different situation arises when the order of the doped wafers is changed, i.e., wafer “1” is placed closer to the incident wave; see expression (5). The absorption data for this case are shown in Figure 4a. Here, we observe a splitting of the resonant absorption peak into two sub-peaks, forming a wider absorption bandwidth. Analysis of the field modulus distribution in Figure 4b shows that the dielectric layers are better-coupled. In particular, for the first sub-peak, the resonance condition is found to be distributed over the whole region $a+d_{1}+a$. 

In addition, the value of the field modulus in the left and right dielectric resonators is comparable since wafer “1” provides less signal attenuation than “2”. Consequently, in this case, the contribution of wafer “2”, located at the end of the structure concerning the incident radiation, to the total absorption becomes essential. At this point, the analogy with the splitting of levels in a double-well potential in quantum mechanics is worth mentioning. Despite the fact that in the situation considered below, there is no barrier structure (the real part of the dielectric permittivity for all layers, including the doped ones, turns out to be positive), an analogue of sub-barrier tunnelling is attenuation associated with signal absorption. In quantum mechanics, tunnelling determines the coupling of wells and, hence, the splitting of planes. In our case, the process of field penetration from the right to the left dielectric plate is responsible for the “coupling” in the heterostructure. This fact partially explains the difference in the formation of absorption zones when the order of the absorbing layers is changed.

Thus, the optimal configuration of periodic heterostructures for realizing efficient signal absorption is primarily determined by the characteristics of the incident radiation. For detecting a quasimonochromatic signal, it is reasonable to use a two-layer structure of type (3), where the absorption is concentrated in a single conducting layer. The multilayer structure can be used to detect broadband signals, but in this case, the doping levels of the conducting layers should be lower than in the case of a two-layer structure. This will ensure that a sufficient fraction of the signal penetrates into the layers far from the incident radiation, resulting in the formation of a wide absorption zone. It should be noted that due to the greater number of signal re-reflections from the conductive layers, the total fraction of absorbed signal may be smaller than in the two-layer structure.

Our experimental and theoretical data have shown that using the proposed anti-reflection coating increases the signal absorption efficiency in the bolometer’s conductive layer. Multilayer structures with finite thicknesses can work similarly on the nanometer scale as microscale heterostructures when the dielectric constant and thickness of the outside layer are tuned appropriately. Bolometers with a Josephson nanostructure integrated into the waveguide are used to detect radiation with record sensitivity at a frequency of several GHz [39,40]. By covering such a structure with our anti-reflection coating, we can improve the efficiency of such a Josephson detector. The whole device can be made compactly, similarly to applying an anti-reflection coating for silicon optics at terahertz frequencies [56,57,58]. Our waveguide seamlessly fits into standard microwave setups for measurements involving superconducting resonators, qubits, and parametric amplifiers [59,60]. In this configuration, the microwave generator is the source of microwave photons. At the same time, a vector network analyzer functions both as a generator and detector of the probe signal (see Figure 5a). The total attenuation of the input line is based on the specific experiment but is typically −80 dB so as to achieve quasi-single phone operation. The transmitted output signal from the waveguide port is amplified by a broadband low temperature amplifier with an average gain of 30 dB. The waveguide (Figure 5b) is typically placed in a well-shielded environment at the base temperature. Furthermore, our anti-reflection coating can serve as additional adsorbing layers in a bolometer based on the Josephson junction Al-AlOx-Al [39,40]. Figure 5c shows a SEM image of a part of the test chip with such a Josephson junction. Our calculations affirm that the appropriate choice of the multilayer bolometric structure’s thickness finds application in cutting-edge nano- and quantum technologies. However, this requires high-quality metal structures with high dielectric permittivity [61,62].

## 4. Conclusions

In summary, our investigation delved into the absorption of GHz signals in structures comprising a single conducting layer in conjunction with a dielectric plate or a pair of such layers, both theoretically and experimentally. Notably, we observed the presence of resonance absorption, characterized by broadening and the emergence of a split absorption peak in the double-layer structure.

The proposed technique enables wideband signal reception. The proposed structure, with only four layers, covers the entire n77 band of 5G NR, from 3300 to 4200 MHz. Note that the signals received by quantum bits based on Josephson nanostructures tend to be in the same frequency range. The experimentally confirmed non-uniformity of the frequency response for this band is 1 dB, which is sufficient for communications applications. The inclusion of additional layers further extends the efficient frequency band. The proposed dielectric–semiconductor flip increases the overall efficiency enough to compensate for the electromagnetic energy loss in numerous layers.

Thus, the proposed structure increases the frequency bandwidth while compensating for the loss in efficiency due to the absorption of the additional layers. The resulting sensor can have quite good bandwidth and efficiency, comparable to the resonant one.

## Figures and Tables

**Figure 1 nanomaterials-14-00141-f001:**
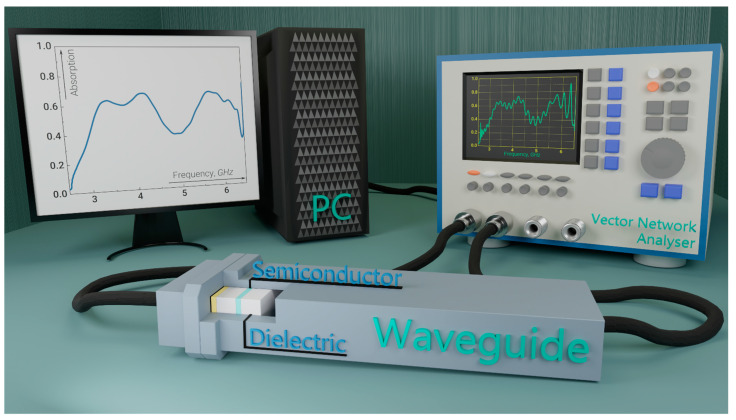
An experimental setup with a closed waveguide system, a signal source and detector, and test samples.

**Figure 2 nanomaterials-14-00141-f002:**
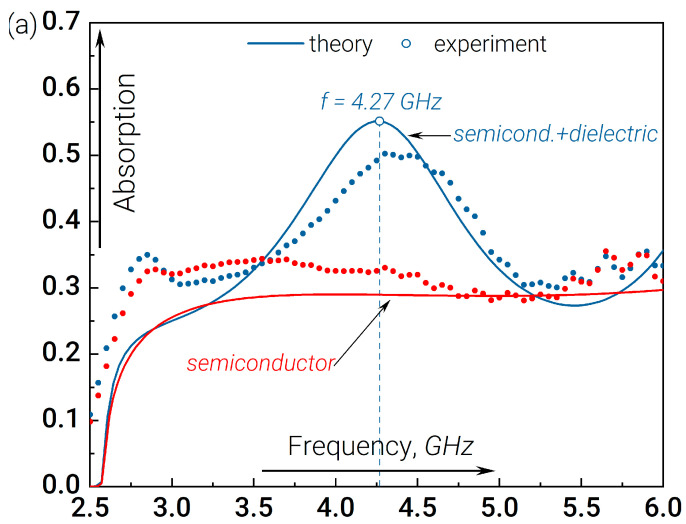
(**a**) Experimental (dots) and theoretical (curves) data on the absorbed GHz radiation normalized to the incident radiation flux measured at the output of the waveguide in the presence of either a bolometric or our resonant structure; see Equation (3). (**b**) The field modulus distribution along the waveguide for the resonant frequency f≈4.3 GHz in the case with/without the dielectric plate. The electric field strength is normalized to the amplitude of the incident wave. For 100% reflection from the structure, the maximum value of the electric field modulus is two.

**Figure 3 nanomaterials-14-00141-f003:**
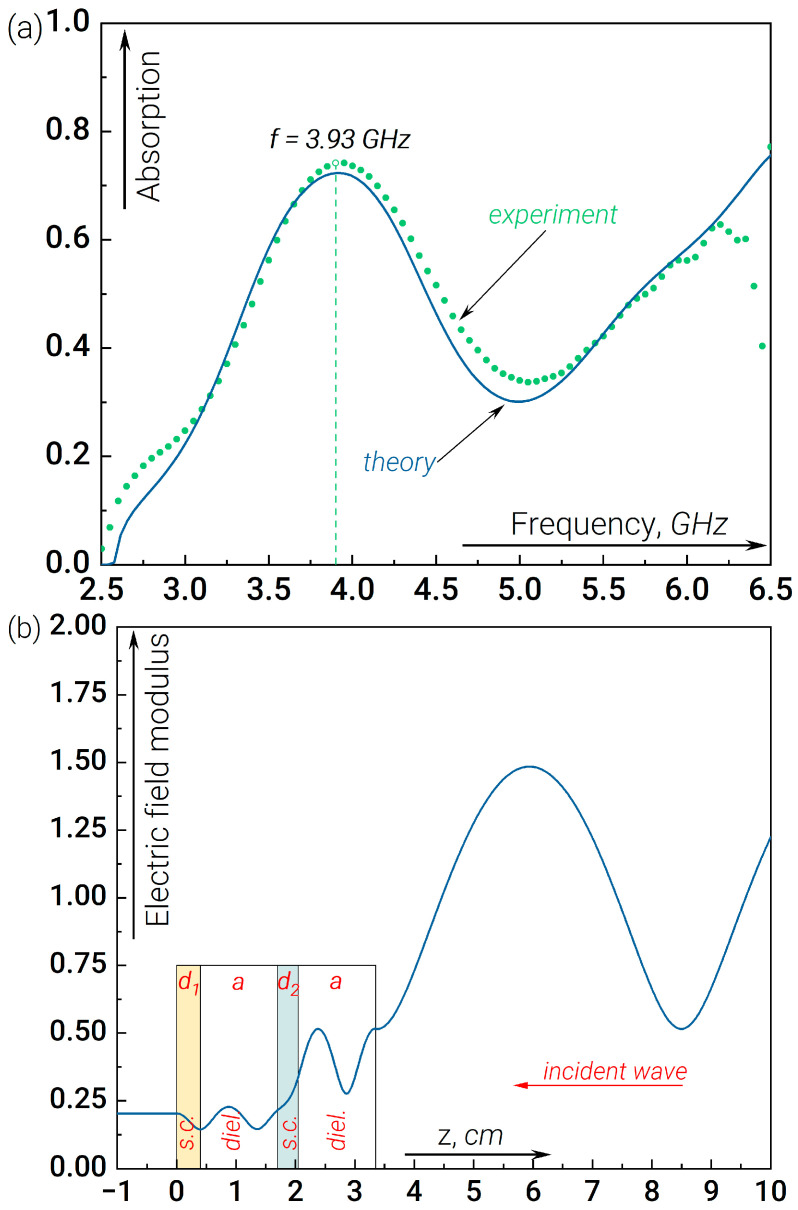
(**a**) Experimental (dots) and theoretical (curves) data on the absorbed GHz radiation normalized to the incident radiation flux measured at the output of the waveguide in the presence of a resonant structure (4) versus the linear signal frequency. (**b**) The field modulus distribution along the waveguide for the resonant frequency f=3.93 GHz. The electric field strength is normalized to the amplitude of the incoming wave.

**Figure 4 nanomaterials-14-00141-f004:**
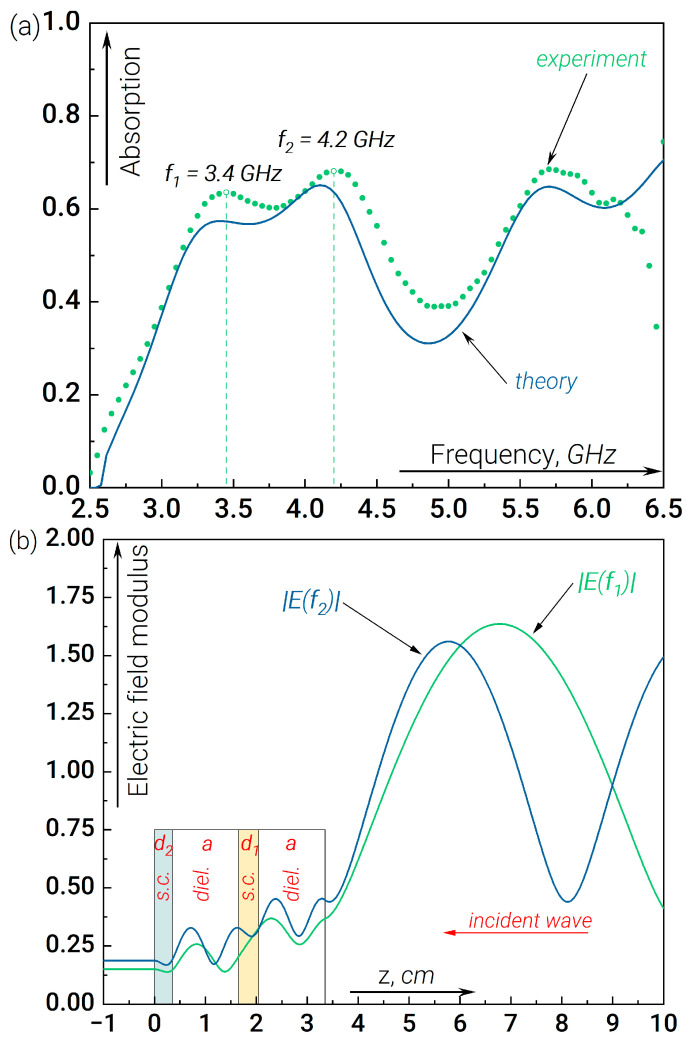
(**a**) Experimental and theoretical data of the absorbed GHz radiation normalized to the incident radiation flux measured at the output of the waveguide in the presence of the double resonant structure “semiconductor + single dielectric” versus the linear signal frequency. The resonant absorption peaks are split into two sub-peaks. The splitting of the peak with the position close to 6 GHz is not pronounced due to the appearance of the multimode waveguide regime. (**b**) The field modulus distribution along the waveguide for *f* = 3.4 GHz (the first sub-peak, green curve) and *f* = 4.2 GHz (the second sub-peak, blue curve). The electric field strength is normalized to the amplitude of the incoming wave.

**Figure 5 nanomaterials-14-00141-f005:**
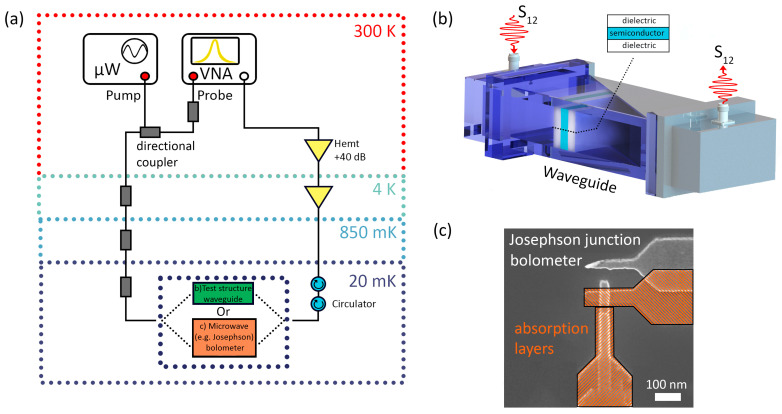
(**a**) The proposed measurement scheme for mK temperatures. (**b**) Sketch of a cryogenic waveguide that will be used to test the results obtained at low temperatures. (**c**) An example of a key part (superconductor–insulator–superconductor tunnel contact) of a Josephson microwave photon detector whose sensitivity will be enhanced by the use of an anti-reflection coating designed for experimental conditions (with appropriate relatively high dielectric constant, appropriate dimensions, etc.). Here, we have two superconducting electrodes (absorption layers) separated in the overlap region by a thin “tunnel” dielectric layer; the absorption of microwave photons leads to a transition to a new (resistive) state of the entire nanostructure, and the proposed anti-reflection coating will increase the efficiency of the detector’s interaction with the field in this case as well.

## Data Availability

Data are contained within the article.

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
