# Peer review of "Multilayer Bolometric Structures for Efficient Wideband Communication Signal Reception"

_nanomaterials, 2024, doi:10.3390/nano14020141_

Round 1

Reviewer 1 Report

Comments and Suggestions for Authors

Dear authors,

Congratulations for this paper which proposed to achieve a multilayer structure to be used as a receiver of electromagnetic signals of given amplitude and bandwidth. The result is a structure with only four layers, which covers the entire n77 band of 5G NR, from 3300 to 4200 MHz. But, it should be noted that the description of the research conducted in this paper is not consistent with the paper title.

Some suggestions:

Title: It should be revised – The construction “Multilayer Bolometric Sensors” it is not frequently used in the text!

Abstract: The abstract should be revised. Usually, in scientific articles, communications are written in the third person and not in the first person.

Keywords: The keywords are the same as in [41],  but  “nanomaterials for sensing “, “semiconductor plasma”, “; quantum radio reception” are not found in the text.

Text

r.72, r.75: The objective of the paper should be better formulated.

r.86 – we deal ?…

r.99 – we? are dealing …

r.173 – to be corrected

r.179, r.180 – In Fig. 2, and Fig. 3, the values of the absorption and electric field strength have the units of measurement?

r.205 – to be corrected

English – it should be improved.

Comments on the Quality of English Language

English – it should be improved.

Author Response

Dear authors,

Congratulations for this paper which proposed to achieve a multilayer structure to be used as a receiver of electromagnetic signals of given amplitude and bandwidth. The result is a structure with only four layers, which covers the entire n77 band of 5G NR, from 3300 to 4200 MHz. But, it should be noted that the description of the research conducted in this paper is not consistent with the paper title.

We warmly thank the Reviewer for careful reading and understanding of the main challenges and results of our manuscript. As the Reviewer will see, we, point by point, considered his/her criticism and substantially modified the manuscript following his/her valuable remarks. The reviewer will find a specific file submitted along with other documents in which the changes done in the main text are highlighted.

Some suggestions:

1) Title: It should be revised – The construction “Multilayer Bolometric Sensors” it is not frequently used in the text!

Thank you for your valuable comment. The title has been revised according to your comments.

2) Abstract: The abstract should be revised. Usually, in scientific articles, communications are written in the third person and not in the first person.

Based on this criticism, we reorganized the abstract and wrote it in the third person.

3) Keywords: The keywords are the same as in [41],  but  “nanomaterials for sensing “, “semiconductor plasma”, “; quantum radio reception” are not found in the text.

Done! Thanks for the comment. We have corrected the keyword set. 

Text

4) r.72, r.75: The objective of the paper should be better formulated.

Done!Thank you very much for the remark! 

5) r.86 – we deal ?…

Done!  We have fixed this problem.

6) r.99 – we? are dealing …

Done! We have corrected this sentence.

7) r.173 – to be corrected

Done!

8) r.179, r.180 – In Fig. 2, and Fig. 3, the values of the absorption and electric field strength have the units of measurement?

We absolutely agree with the reviewer and apologize for the mistake!

We have added the necessary explanations in the captions to the figures.

9) r.205 – to be corrected

Done

Thank you for your valuable comment.

We have completely revised the conclusions of our article.

Reviewer 2 Report

Comments and Suggestions for Authors

1. The authors' claim (lines 247 - 261, including Fig.5) that "Our experimental data indicates that the potential integration of our anti-reflection coating with a Josephson junction bolometer can lead to heightened sensitivity in a Josephson junction bolometer" requires deeper justification.

2. The description of the suggested "measurement setup for millikelvin 249 temperatures (see Figure 5)" is unclear and requires clarification.

3. In connection with the proposal to apply the results obtained on the periodic heterostructures of the microwave range to nanoscale Josephson junction bolometers, a comment should be made on scaling the microwave heterostructure (dimensions) to the nanometer scale.

4. There are typing errors in the caption of Fig. 5 and in Fig. 5c.

Author Response

  1. The authors' claim (lines 247 - 261, including Fig.5) that "Our experimental data indicates that the potential integration of our anti-reflection coating with a Josephson junction bolometer can lead to heightened sensitivity in a Josephson junction bolometer" requires deeper justification.

We appreciate the comment very much. We rearranged the manuscript according to the Reviewer's requirements. We absolutely agree with the reviewer, we have added a detailed explanation to the text. We have additionally included recent articles on applying antireflection coatings [Miao, Wei, et al. "A terahertz detector based on superconductor-graphene-superconductor Josephson junction." Carbon 202 (2023): 112-117.]. 

2. The description of the suggested "measurement setup for millikelvin 249 temperatures (see Figure 5)" is unclear and requires clarification

Done! Thanks for the comment. We have added an additional description of the measurement setup.

3. In connection with the proposal to apply the results obtained on the periodic heterostructures of the microwave range to nanoscale Josephson junction bolometers, a comment should be made on scaling the microwave heterostructure (dimensions) to the nanometer scale.

We want to thank the Reviewer for this comment! We slightly modified the text.

4. There are typing errors in the caption of Fig. 5 and in Fig. 5c.

Thank you for your valuable comment. We have fixed the problem. 

Reviewer 3 Report

Comments and Suggestions for Authors

The primary focus of the research is to demonstrate the feasibility of receiving broadband electromagnetic signals using a multilayer metamaterial. The study explores the construction of a lattice of resonators made of alternating conducting (Silicon) and dielectric layers to enhance the absorption of electromagnetic energy. The study demonstrates the wideband absorption of gigahertz waves within the proposed multilayer structure. The research addresses the pressing need for expanding communication systems into various frequency bands while mitigating electromagnetic spectrum pollution. By proposing a multilayer structure as a wideband electromagnetic signal receiver, the study introduces an interesting approach to bolometric detectors, filling a gap in achieving higher sensitivity for radiofrequency signals. The manuscript builds upon previous work on resonance effects in bolometric detectors. It extends the concept by introducing a multilayer structure to create broad spectral zones for electromagnetic energy absorption. The addition of several conductive and dielectric layers aims to overcome the limitation of narrow bandwidth associated with resonant detectors. The methodology is sound. The conclusions are generally consistent with the evidence presented. The discussion on the trade-off between bandwidth and efficiency is insightful. The figures effectively illustrate the concept and outcomes of the research. In overall, the manuscript is well-written and well-organized. I recommend its publication after the authors address the following comments:

1. The abstract could benefit from a clearer articulation of the primary findings. Mentioning key numerical results in the abstract could enhance its effectiveness.

2. Did the authors use commercial software for simulations? If so, could you please elaborate on the specific software employed?

3. The references appear relevant and support the context of the study. However, consider providing a brief introduction to metamaterials in the introduction section, along with their applications, such as:

- 10.1364/AO.393501

- 10.1364/JOSAB.446803

4. Consider comparing your results with previous studies in a table, if possible.

Author Response

The primary focus of the research is to demonstrate the feasibility of receiving broadband electromagnetic signals using a multilayer metamaterial. The study explores the construction of a lattice of resonators made of alternating conducting (Silicon) and dielectric layers to enhance the absorption of electromagnetic energy. The study demonstrates the wideband absorption of gigahertz waves within the proposed multilayer structure. The research addresses the pressing need for expanding communication systems into various frequency bands while mitigating electromagnetic spectrum pollution. By proposing a multilayer structure as a wideband electromagnetic signal receiver, the study introduces an interesting approach to bolometric detectors, filling a gap in achieving higher sensitivity for radiofrequency signals. The manuscript builds upon previous work on resonance effects in bolometric detectors. It extends the concept by introducing a multilayer structure to create broad spectral zones for electromagnetic energy absorption. The addition of several conductive and dielectric layers aims to overcome the limitation of narrow bandwidth associated with resonant detectors. The methodology is sound. The conclusions are generally consistent with the evidence presented. The discussion on the trade-off between bandwidth and efficiency is insightful. The figures effectively illustrate the concept and outcomes of the research. In overall, the manuscript is well-written and well-organized. I recommend its publication after the authors address the following comments:

We appreciate the comment very much. We rearranged the manuscript according to the Reviewer requirements.

  1. The abstract could benefit from a clearer articulation of the primary findings. Mentioning key numerical results in the abstract could enhance its effectiveness.

We warmly thank the Reviewer for careful reading and understanding of the main challenges and results of our manuscript. The abstract has been modified based on your comments.

2. Did the authors use commercial software for simulations? If so, could you please elaborate on the specific software employed?

We mentioned the software used in the text.

3. The references appear relevant and support the context of the study. However, consider providing a brief introduction to metamaterials in the introduction section, along with their applications, such as:

  • 10.1364/AO.393501

    - 10.1364/JOSAB.446803

Done! Thanks for the comment. 

Thank you for your valuable comment.

We have completely revised the conclusions of our article.